# Low-Temperature Pyrolysis of Municipal Solid Waste Components and Refuse-Derived Fuel—Process Efficiency and Fuel Properties of Carbonized Solid Fuel

**Kacper Świechowski** [1,*] [iD]**, Ewa Syguła** [1] [iD]**, Jacek A. Koziel** [2] [iD]**, Paweł Stępień** [1] [iD]**, Szymon Kugler** [3]**, Piotr Manczarski** [4] [iD] **and Andrzej Białowiec** [1,2] [iD]

[1] Faculty of Life Sciences and Technology, Institute of Agricultural Engineering, Wrocław University of Environmental and Life Sciences, 37/41 Chełmońskiego Str., 51-630 Wrocław, Poland; ewa.barbara.sygula@gmail.com (E.S.); pawel.stepien@upwr.edu.pl (P.S.); andrzej.bialowiec@upwr.edu.pl (A.B.)

[2] Department of Agricultural and Biosystems Engineering, Iowa State University, Ames, IA 50011, USA; koziel@iastate.edu

[3] Polymer Institute, Faculty of Chemical Technology and Engineering, West Pomeranian University of Technology, 10 Pułaskiego Str., 70-322 Szczecin, Poland; szymon.kugler@zut.edu.pl

[4] Department of Environmental Engineering, Hydro and Environmental Engineering, Faculty of Building Services, Warsaw University of Technology, 00-661 Warszawa, Poland; piotr.manczarski@pw.edu.pl

* Correspondence: kacper.swiechowski@upwr.edu.pl

**Abstract:** New technologies to valorize refuse-derived fuels (RDFs) will be required in the near future due to emerging trends of (1) the cement industry's demands for high-quality alternative fuels and (2) the decreasing calorific value of the fuels derived from municipal solid waste (MSW) and currently used in cement/incineration plants. Low-temperature pyrolysis can increase the calorific value of processed material, leading to the production of value-added carbonized solid fuel (CSF). This dataset summarizes the key properties of MSW-derived CSF. Pyrolysis experiments were completed using eight types of organic waste and their two RDF mixtures. Organic waste represented common morphological groups of MSW, i.e., cartons, fabrics, kitchen waste, paper, plastic, rubber, PAP/AL/PE composite packaging (multi-material packaging also known as Tetra Pak cartons), and wood. The pyrolysis was conducted at temperatures ranging from 300 to 500 °C (20 °C intervals), with a retention (process) time of 20 to 60 min (20 min intervals). The mass yield, energy densification ratio, and energy yield were determined to characterize the pyrolysis process efficiency. The raw materials and produced CSF were tested with proximate analyses (moisture content, organic matter content, ash content, and combustible part content) and with ultimate analyses (elemental composition C, H, N, S) and high heating value (HHV). Additionally, differential scanning calorimetry (DSC) and thermogravimetric analyses (TGA) of the pyrolysis process were performed. The dataset documents the changes in fuel properties of RDF resulting from low-temperature pyrolysis as a function of the pyrolysis conditions and feedstock type. The greatest HHV improvements were observed for fabrics (up to 65%), PAP/AL/PE composite packaging (up to 56%), and wood (up to 46%).

**Dataset:** The dataset is submitted as Supplementary Material to this paper.

**Keywords:** refuse-derived fuel; pyrolysis; carbonized solid fuel; high heating value; waste-to-energy; waste management; sustainability; circular economy; waste-to-carbon

## 1. Summary

Incineration is one of the main treatments available for municipal solid waste (MSW). Incineration largely reduces the mass of waste, eliminating many negative aspects of landfilling. Incineration can also be an advantageous way to dispose of toxic MSW. More often, MSW is used in an energy recovery process to produce electricity and heat. Cement plants can use MSW for energy recovery and material recycling, and it can also be used to generate refuse-derived fuel (RDF). The name "RDF" is mainly applied when enhancing resource recovery from MSW. The conversion of MSW to RDF is a priority in industrialized countries, where RDF is considered as a high-energetic-value material with homogeneous particle size [1–3].

RDF, an alternative fuel, is produced from a mechanically separated oversize fraction produced in a mechanical–biological–municipal waste treatment plant (MBT). The RDF can be produced in an MBT, but also in dedicated installations for producing alternative fuels. In dedicated plants, RDFs are produced from pre-RDF (a raw oversize MSW fraction) or another residual waste [4]. Usually, the production of RDF consists of the separation of high energetic fractions from other waste negatively affecting the quality of RDF. The negative effect on RDF quality is caused by materials that have a low calorific value, e.g., organic waste, wet paper, etc., or inert materials such as glasses, ceramics, ashes, and so on. Waste containing heavy metals and chlorine has a negative impact on RDF quality and environmental safety due to harmful emissions during combustion and excessive equipment wear [5]. Metals and chlorine can be removed in the MBT, and the water content in the organic waste can be decreased by prior bio-drying or a conventional drying process [6].

Today, alternative fuels are an important source of energy for cement plants, known to be large consumers of energy. Approximately 80% of cement installations in Europe incinerate/co-incinerate alternative fuels. RDF produced from an oversize fraction of MSW is the main alternative source of energy used in cement plants in Poland [2]. Moreover, Poland cement plants also use other alternative fuels, e.g., waste rubber, worn tires, waste oils, solvents, textiles, dried sludge, animal waste, and other specially prepared waste mixtures [7]. In 2018 alone, around 2.8 million Mg of MSW was prepared for incineration/incinerated in order to recover energy in Poland [8], and up to 1.5 million Mg of this stream could be used in cement plants [9]. Despite the incineration of a large amount of residual waste, there is a problem of overproduction of RDF in Poland [10]. This problem partially results from the law prohibiting landfilling of waste with a calorific value higher than 6 MJ·kg$^{-1}$. As a result, self-ignition often occurs during long and incorrect storage.

In recent times, the European Union has started the transition towards a circular economy in which waste management has an important role. The key challenge of this transition is to change the perception of "waste as a problem" to "waste as a resource". This situation is reflected by established targets in waste reuse and recycling for E.U. countries. By 2030, landfilling of MSW has to be reduced up to 10%, and 65% of MSW has to be prepared for reuse or recycling [11]. This will require increased efficiency of separating raw material waste (e.g., plastics, cardboard) for recycling. The result of this will be a decrease in the calorific value of residual waste, negatively influencing the calorific value of RDF. This second effect of increasing MSW recycling could force cement/incineration plants to seek better-quality alternative fuels. The E.U. 2017 document, "The role of waste-to-energy in the circular economy", states that incineration plants will not be promoted directly. However, they will continue to be an important element of waste management in the path to a circular economy, disposing of non-recyclable waste [12]. For this reason, incineration plants have to be provided with fuel that will allow them to work efficiently, despite the deterioration in quality of the raw material used in RDF production.

Thus, owing to the tendencies of (i) increases in demand for high-quality alternative fuel by the cement industry, (ii) overproduction of oversize fractions that cannot be handled in another way, and (iii) the falling calorific value of RDF as a result of increased recycling efforts, methods of RDF upgrade will be needed in the near future. One of these methods is low-temperature pyrolysis. In general, the pyrolysis of waste requires additional and external sources of energy. However, it also

has advantages. Pyrolysis results in an increase in the calorific value (similar to bitumen coal) and a decrease in the amount of waste. Pyrolysis is a way to make RDF more homogeneous, safer for storage, and less costly to transport due to energy densification and mass waste reduction.

Pyrolysis is a process of thermal degradation of organic matter in the absence of oxygen which produces char (in this case, carbonized solid fuel CSF), oil, and gas. In the case of low-temperature pyrolysis, the temperatures used are from 400 °C to 550 °C [13–15]. Pyrolysis is regarded, alongside incineration in cement plants, as the future of thermal waste recovery [16]. Nevertheless, there are some challenges with waste pyrolysis. The high heterogeneity of MSW composition (which depends on many factors, e.g., place of origin, season of the year, weather, development status, etc.) makes the RDF composition unstable [17], and various components of RDF may react independently during pyrolysis [14,16]. For this reason, it is challenging to control the RDF composition, which then creates problems with proper process operation and control of the fuel properties of CSF.

Due to the lack of research on low-temperature pyrolysis for specific morphological groups of MSW, their impact on the CSF's chemical and physical properties has not yet been characterized. To date, the thermal transformation of particular waste groups is also unknown. The data contained in the dataset of this work may be used to determine the physical, chemical, and fuel properties of CSF produced from particular organic solid waste. The results can also be used to process kinetic determination and to model the energy balance of low-temperature pyrolysis of particular waste, e.g., in accordance with [18–20].

## 2. Data Description

### 2.1. The Origin of Materials

In this study, eight types of common organic waste and two RDF mixtures were prepared and examined. In total, ten materials were used. Each type of organic waste represented a morphological group of MSW. The organic waste groups were cartons, fabrics, kitchen waste, paper, plastic, rubber, PAP/AL/PE composite packaging (multi-material packaging also known as Tetra Pak cartons), and wood. The particular groups of waste were represented by materials according to Table 1. A mixture of kitchen waste was prepared in accordance with Yang et al. 2013 [21].

**Table 1.** Selected research materials assigned to morphological waste groups.

| Morphological Group of Waste | Material |
| --- | --- |
| Carton | Grey carton |
| Fabrics | Cotton t-shirt |
| Kitchen waste | Vegetables 41.6% (carrot 13.86%, potato 13.86%, salad 13.86%), banana peel 29.7%, basic food (pasta 7.43%, rice 7.43%, bread 7.43%), chicken 0.2%, eggshells 4%, and walnut shells 2.2% by weight |
| Paper | Office paper |
| Plastics | Polyethylene foil |
| Rubber | Car inner tube |
| PAP/AL/PE composite packaging | Tetra Pak packaging |
| Wood | Branches from pruning of trees |

The two RDF blends were prepared from the morphological groups of waste used (Table 1). The composition of the RDF blends was selected in accordance with Stępień et al. 2018 [22], and their weight percentage shares are presented in Table 2. The RDF presented by Stępień et al. 2018 [22] also contained unidentified waste, the share of which was up to ~30%. In this study, that share of the unidentified fraction was divided into the number of blend constituents, and the result was added to a particular constituent, so the sum of the presented RDF blends is 100% (Table 2).

**Table 2.** Composition of prepared refuse-derived fuel (RDF) blends.

| Morphological Group of Waste | RDF-1, % | RDF-2, % |
|---|---|---|
| Carton | 9.64 | 8.57 |
| Fabrics | 6.20 | 9.54 |
| Kitchen waste | 4.02 | 7.10 |
| Paper | 9.64 | 8.57 |
| Plastics | 34.23 | 45.24 |
| Rubber | 9.60 | 7.71 |
| PAP/AL/PE composite packaging | 12.22 | 5.81 |
| Wood | 14.45 | 7.46 |

### *2.2. The Properties of Raw and Pyrolyzed Materials*

The prepared samples were used for CSF production. Then, the raw material samples and produced CSF were tested via proximate and ultimate analyses and for high heating value (*HHV*). The pyrolysis process efficiency was described by parameters such as mass yield, energy densification ratio, and energy yield. The raw materials were also analyzed by thermogravimetric analysis (TGA) and differential scanning calorimetry (DSC). The obtained data are presented in the Supplementary Material file "Low-temperature pyrolysis of organic waste.xlsx" in seven sheets that are organized as follows:

- Read-me (guide);
- Pyrolysis process;
- Proximate analysis;
- HHV;
- Ultimate analysis;
- DSC;
- TGA.

The "Read-me" spreadsheet is about how to read the data. The second spreadsheet ("Pyrolysis process") contains data on the pyrolysis mass yield, energy densification ratio, and energy yield of CSF. The third spreadsheet ("Proximate analyses") provides data on the moisture content, organic matter content (understood as lost on ignition), ash content, and combustible parts in the materials and CSFs. The fourth spreadsheet ("HHV") contains data on the high heating value of the materials and CSFs. The fifth spreadsheet ("Ultimate analyses") presents the C, H, N, and S composition of the materials and CSFs. The sixth spreadsheet ("DSC") presents the DSC results, and the seventh spreadsheet ("TGA") presents the TGA results.

## 3. Methods

### *3.1. Material Preparation*

For each waste material, around 1 kg of the primary sample was collected/prepared. The primary samples were dried, milled, and then stored in a freezer at −15 °C before tests. The drying process was conducted in a laboratory dryer (WAMED, model KBC-65W, Warsaw, Poland) at 105 °C for at least 24 h. The materials were considered "dry" when the difference in mass during drying did not change over 1 h. Next, dry materials were ground through a 1 mm screen using a laboratory knife mill (Testchem, model LMN-100, Pszów, Poland). Before a specific test, samples were defrosted to room temperature and then once again dried in a laboratory dryer at 105 °C for 24 h.

### *3.2. Carbonized Solid Fuel (CSF) Production*

The pyrolysis leading to the production of CSF was performed in a muffle furnace (Snol 8.1/1100, Utena, Lithuania) according to the methodology presented in previous work [23]. For CSF generation,

$CO_2$ inert gas was used with a flow rate of ~2.5 $dm^3 \cdot min^{-1}$, and the gas was introduced to the middle of the reactor. CSF was produced under 300–500 °C with intervals of 20 °C, during residence times of 20–60 min with intervals of 20 min. The dry samples (10 ± 0.5 g) of materials in an inert ($CO_2$) atmosphere were heated from room temperature (20 °C) to the low-temperature pyrolysis setpoint temperature at a heating rate of 50 °C$\cdot min^{-1}$. The muffle furnace started work 5 min after the inert gas started to flow to the inside of the muffle furnace. After the process (when the temperature dropped below 200 °C), the CSF samples were removed from the muffle furnace. After that, the weight loss and process mass yield were measured and estimated based on the mass before and after the process. A laboratory balance (Radwag, model As 220. R2, Radom, Poland) with accuracy of 0.1 g was used. The produced samples of CSF were stored in plastic containers in room conditions, temperature ~20 °C, for further tests. A graph of process temperature versus process time for the 500 °C setpoint is presented in Figure 1. Figure 1 shows the temperature change during the cooling process. The graphic can be used to estimate the cooling time for each pair of temperature setpoint and process residence time. Red lines mark the assumed end of the process.

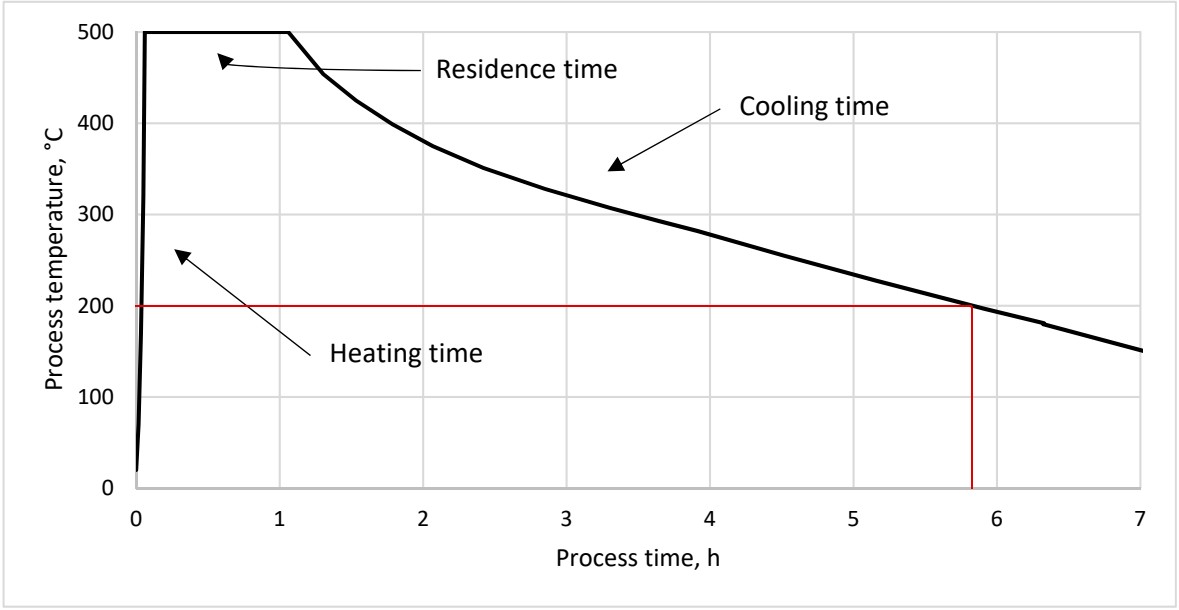

**Figure 1.** An example of temperature patterns during the pyrolysis of municipal solid waste (MSW) components.

The mass yield (*MY*), energy densification ratio (*EDr*), and energy yield (*E.Y.*) of CSF production were determined based on Equations (1)–(3), respectively.

$$Mass\ yield = \frac{\text{mass of raw material before the pyrolysis process}}{\text{mass of CSF after the pyrolysis process}} \cdot 100 \tag{1}$$

$$Energy\ densification\ ratio = \frac{\text{high heating value of CSF}}{\text{high heating value of raw material}} \tag{2}$$

$$Energy\ yield = Mass\ yield \cdot energy\ densification\ ratio \tag{3}$$

### 3.3. Proximate and Ultimate Analysis of Raw and Carbonized Materials

Proximate analyses of the raw materials and carbonized solid fuels were performed using the following equipment and standards:

- The moisture content (*MC*) was determined using a dryer (WAMED, model KBC-65W, Warsaw, Poland) according to the PN-EN 14346:2011 standard [24], in three replications;

- The organic matter content measured as loss on ignition (*O.M.*) was determined using a muffle furnace (Snol 8.1/1100, Utena, Lithuania) according to the PN-EN 15169:2011 standard [25], in three replications;
- The ash and combustible part (*C.P.*) was determined using a muffle furnace (Snol 8.1/1100, Utena, Lithuania) according to the PN-Z-15008-04:1993 standard [26], in three replications;
- The *HHV* was determined using a calorimeter (IKA$^{®}$ Werke GmbH, model C200, Staufen, Germany) according to PN-Z-15008-04:1993 [27], in two replications. During the experiment, for all CSF samples, with the exclusion of PAP/AL/PE composite packaging, the mass was 0.2 ± 0.05 g. For CSF made from PAP/AL/PE composite packaging, the mass of samples for testing was reduced to 0.01 g ± 0.002 g. The reason for that was the high content of aluminum in the PAP/AL/PE composite packaging, which did not oxidize completely for a larger sample mass.

The elemental compositions (C, H, N, and S) of materials and CSFs were determined using a CHNS (Perkin Elmer, model 2400 CHNS/O Series II System) analyzer in accordance with PKN-ISO/TS 12902:2007 [28], in one replication.

### 3.4. Thermogravimetric Analysis (TGA) of Raw Materials

Thermogravimetric analysis was done in non-isothermal conditions. The materials were heated from 20 to 500 °C at three heating rates. The heating rates were 5 °C·min$^{-1}$, 10 °C·min$^{-1}$, and 15 °C·min$^{-1}$. For each heating rate, three replications were done. The analysis of the thermal degradation was performed using a TGA analyzer (Mettler Toledo, TGA 2, Schwerzenbach, Switzerland). Samples of 4 mg were used. Nitrogen was used as an inert gas, with a flow rate of 3 dm$^3$·h$^{-1}$.

### 3.5. Differential Scanning Calorimetry Analysis of Raw Materials

For differential scanning calorimetry analysis (DSC), a scanning calorimeter (TA Instruments, DSC Q2500, New Castle, DE, USA) was used. The ~6 mg raw material samples were prepared and placed in the aluminum hermetic crucible. The aluminum crucible was placed into the analyzer, where it was heated from 20 to 550 °C. The heating rate was 10 °C·min$^{-1}$, and the flow rate of nitrogen was 3 dm$^3$·h$^{-1}$. One replication was carried out for each material [29].

## 4. User Notes

This work presents the results of the low-temperature pyrolysis of different morphological components of MSW and produced RDFs. The Supplementary Material contains the mass yield, energy densification ratio, and energy yield results of the pyrolysis process. The data contained in the Supplementary Material also present the main fuel properties of carbonized soil fuels made from different components of MSW and RDFs. These data can be used in the preparation of empirical models to describe the influence of temperature, residence time, and the composition of the feedstock on fuel properties for different groups of organic waste derived from MSW. Based on the presented data, it is possible to initially decide which group of organic waste should be excluded and which fraction should be a significant component when CSF is produced. The presented results are also valuable data for comparison analysis with other types of energy sources, e.g., energy crops. The presented TGA and DSC data can be used to determine the pyrolysis process kinetics and energy demand of the pyrolysis process.

**Supplementary Materials:** The following are available online at http://www.mdpi.com/2306-5729/5/2/48/s1.

**Author Contributions:** Conceptualization, K.Ś. and P.S.; methodology, A.B., K.Ś., E.S., and P.S.; validation, A.B.; investigation: E.S.—biochar generation and proximate analysis, K.Ś.—high heating value determination, P.S.—raw sample preparation, ultimate analysis, and thermogravimetric analysis, S.K.—differential scanning calorimetry; resources, P.S.; data curation, K.Ś.; writing—original draft preparation, K.Ś., P.S.; writing—review and editing,

K.Ś., E.S., A.B., J.A.K.; visualization, K.Ś.; supervision, A.B., J.A.K., P.M.; project administration, P.S.; funding acquisition, P.S. All authors have read and agreed to the published version of the manuscript.

**Funding:** The research was supported by the Preludium 14 Program, the National Science Centre, Poland, grant UMO-2017/27/N/ST8/03103 titled "Energy balance of low temperature pyrolysis of organic waste". This research was partially supported by the Iowa Agriculture and Home Economics Experiment Station, Ames, Iowa. Project no. IOW05556 (Future Challenges in Animal Production Systems: Seeking Solutions through Focused Facilitation) sponsored by Hatch Act & State of Iowa funds.

**Acknowledgments:** The presented test results were obtained as part of the activity of the leading research team - Waste and Biomass Valorization Group (WBVG).

**Conflicts of Interest:** The authors declare no conflict of interest. The funders had no role in the design of the study; in the collection, analyses, or interpretation of data; in the writing of the manuscript; or in the decision to publish the results.

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
