# Peer review of "Low-Temperature Pyrolysis of Municipal Solid Waste Components and Refuse-Derived Fuel—Process Efficiency and Fuel Properties of Carbonized Solid Fuel"

_data, 2020_

Round 1

Reviewer 1 Report

see attached file

Reviewer 2 Report

This preliminary pyrolysis also requires energy – there is no matter if the process proceeds in a separate unit (furnace). Maybe this is even worse. So where is the advantage ? Please comment this problem.

lines 58-60 and 76        Please do not use symbols HHV (higher heating value) and LHV (lower heating value) for  personal meaning! It seems to be a misunderstanding… Generally, the sentence is incomprehensible …

line 67       “floatation” ?

line 136     “Reed” – read ?

lines 131-150    This fragment should be the final chapter n a form of “results” not “properties”. Therefore, the chapter 2.2 may be deleted.

lines 153-156    This is a partial repetition of 122-126.

line 160     You used CO2 assuming that the Boudouard reaction (CO2+C=2CO) did not proceed at 500 degrees ? Have you any evidence ? The equilibrium constant for this reaction = 4.1e-3 at 500 Celsius degrees. Why you did not use the N2 ? What would be an error if the equilibrium had been achieved ? Please compare HHV for “plastic” > 460 degrees in waste.xlsx and general decrease of HHV for other materials !

line 225     Are you sure that it was “aluminum” ?

line 229     “pioneering” ? Please be a bit modest …

Reviewer 3 Report

This paper presents detailed empirical data from studies of low-temperature pyrolysis of a wide variety of feedstocks generated from municipal solid waste. I believe the paper provides a meaningful contribution to literature, and in fact it will be very useful for my own research in biomass pyrolysis and biochar. The following comments are provided as suggested minor revisions to improve the quality of the final manuscript and data file:

  • Manuscript:
    • Section 3.2: A heating rate of 50°C∙min-1 seems high relative to many prior literature studies and the process we use in our laboratory. Please comment on the selection of this heating rate value, and what effect this may have had on the measured char properties.
    • Why was CO2 selected as the inert gas for CSF production (Section 3.2), but N2 was used for TGA (Section 3.4) and DSC (Section 3.5)?
    • Check throughout that the reference numbers are correct. For example, in Lines 115 and 116, Stępień et al. 2018 [15] should be [16].
    • Lines 63/64: Change “Today alternative fuels are an important source of energy for cement plants. Cement plants are large consumers of energy.” to “Today alternative fuels are an important source of energy for cement plants, known to be large consumers of energy.”
    • Line 79: Need closed quotation mark after “….promoted directly.”
    • Line 94: change “instable” to “unstable”
    • Line 136: Change “Reed me (guide);” to “Read me (guide);”
  • Supplementary Material data file:
    • For all worksheets, where “%” is used in the column heading, there is no need to have “%” with each number in the table.
    • In the Pyrolysis process and Ultimate analysis worksheets it is stated that the analysis has been done in one replication for each material. Please comment on how it was determined that single samples are sufficient (as opposed to performing measurements in duplicate or triplicate), especially with mixed materials that may be expected to have some degree of heterogeneity. Why were these procedures different than Proximate analysis (three replications) and HHV (two replications)?

Reviewer 4 Report

Comments on data-805284;

The manuscript titled “Low-temperature Pyrolysis of Municipal Solid Waste Components and Refuse Derived Fuel – Process Efficiency and Fuel Properties of Carbonized Solid Fuel” by K. Świechowski et al. summarized the key properties of municipal solid waste-derived carbonized solid fuel. The authors provided important data to determine the physical, chemical, and fuel properties of carbonized solid fuels. Since finding alternative fuel sources is a very important task, this work should provide useful insight to those working in the field. I recommend acceptance for data with a present form.
